# KFC: A Scalable Approximation Algorithm for $k$-center Fair Clustering *

**Elfarouk Harb***
Independent Researcher
Hong Kong
eyfmharb@gmail.com

**Lam Ho Shan***
Hong Kong University of Science and Technology
Hong Kong
hslamaa@connect.ust.hk

## Abstract

In this paper, we study the problem of fair clustering on the $k-$center objective. In fair clustering, the input is $N$ points, each belonging to at least one of $l$ protected groups, e.g. male, female, Asian, Hispanic. The objective is to cluster the $N$ points into $k$ clusters to minimize a classical clustering objective function. However, there is an additional constraint that each cluster needs to be fair, under some notion of fairness. This ensures that no group is either "over-represented" or "under-represented" in any cluster. Our work builds on the work of Chierichetti et al. (NIPS 2017), Bera et al. (NeurIPS 2019), Ahmadian et al. (KDD 2019), and Bercea et al. (APPROX 2019). We obtain a randomized $3-$approximation algorithm for the $k-$center objective function, beating the previous state of the art ($4-$approximation). We test our algorithm on real datasets, and show that our algorithm is effective in finding good clusters without over-representation or under-representation, surpassing the current state of the art in runtime speed, clustering cost, while achieving similar fairness violations.

## 1 Introduction and Prior Work

The aim of classical clustering is to group $N$ points into $k$ clusters in order to minimize an objective function, such as $k - \{$centers, medians, means, $...\}$. The problem has been studied extensively and there is a plethora of papers discussing approximation algorithms and PTAS algorithms for different objective functions. (See [8, 4] for $k$-median, [13, 15, 16] for $k-$center, [18, 20] for $k$-means, and [1] for a recent survey).

Variants of clustering problems have been proposed to impose additional constraints to attain a desired demographic in clustering outcome, diversity of a cluster for instance. The constraint of interest in this paper is fairness, it is motivated by practical scenarios encountered when formulating clustering problems that require fairness and has received a lot of attention recently [2, 5, 9, 7, 11, 12, 17, 10]. Ahmadian et al. [2] provided some examples such as clustering news articles while requiring that no one view point or news source over-represents any cluster. They also noted its necessity in online advertising systems to ensure that no advertiser has the majority market share over certain keyword clusters. Other examples where a fair clustering is crucial include clustering people to choose who is awarded loans [19, 21], or predicting recidivism [3] where using classical clustering techniques can be detrimental to minority groups. We discuss this with more details in the broader impact section.

There are several definitions of fairness in the literature. Dwork. et al. [11] used the Lipschitz inequality, by defining a clustering $M$ as a map that maps a point in the inputs, to a probability

distribution over $k$ clusters. The objective was to find a clustering mapping $M$ that minimizes some utility loss function, with $M$ subject to the Lipschwitz inequality as a notion of fairness. Other papers [12, 17] tried to impose fairness by either eliminating disparate impact from the resulting clusters, or using regularization techniques that attempt to remove prejudice, which was shown to be effective when being used on logistic regression.

In this paper, we focus on fairness, or equivalently the "over/under-representation" constraint, and restrict ourself to the $k-$center objective function. Firstly, the classical $k-$center problem is NP-hard and permits a $2-$approximation [13, 15]. Meanwhile, finding a better approximation ratio than 2 is also NP-hard [16]. Several approximation algorithms for fair clustering have been proposed in recent years. Chierichetti et al. [9] extended the definition of disparate impact to clustering problems. Assuming each point in the input has a "color" (representing its group), they defined fair cluster as a cluster where the distribution of colors in each cluster is the same as the distribution of colors over all points. They gave the idea of fairlets to provide an approximation algorithm to the problem when there are only 2 groups (e.g. males or females). This was later generalized for multiple groups (e.g white, Asian, Hispanic, ...) by Rösner et al. [24] to a $14-$approximation algorithm for the $k-$center objective. However, their definition of fairness is quite restrictive, as sometimes there are some input instances where such a fair cluster by that definition does not exist.

Finally, the closest work to ours are the papers by Ahmadian. et al. (KDD 2019) [2], Bera. et al. (NeurIPS 2019) [5], and Bercea et al. (APPROX 2019) [6]. Their idea of fairness stems from the classical $k-$center problem. Given a user defined parameter $\alpha$, [2] defined a fair cluster as a cluster where the fraction of points with a given color in any cluster is at most $\alpha$ (known as the restricted dominance constraint). This ensures that no group "dominates" any cluster. However, we note that [2] and [6] did not allow overlapping between groups, e.g. an input point **cannot** belong to both "male" and "Asian" groups. [5] generalized this by allowing groups to overlap. They also allowed each group $g$ to have its own $\alpha_g$, as well as a $\beta_g$ that ensures that the fraction of points belonging to $g$ in any cluster is at least $\beta_g$ (known as the minority protection constraint) and at most $\alpha_g$. Note that [2] formulation is a special case of [5] formulation. [6] also gave an elegant, yet computationally expensive, approximation framework for fair clustering with various objectives including $k-$center.

Currently, the state-of-the-art generalized formulation by [5] is a $5-$approximation linear program (or $4$-approximation for the case when the centers and clients are the same). [2] provided a $3-$approximation for the restricted version of their formulation. In this paper, we provide a new $5-$ approximation algorithm for the generalized formulation (or $3-$ approximation for the same special case as [5], and thus beating the previous $4-$approximation state of the art). We show theoretically and practically that our algorithm is orders of magnitude faster than the state of the art, more cost effective, and has a very small additive violation that is comparable to the state of the art.

## 2  Preliminaries

Let $C$ be a set of $N$ points embedded in a metric space $(\mathcal{X}, d)$, and $F \subset \mathcal{X}$ be a set of potential cluster centers (i.e "facilities"). Note that in our analysis, we show that we can strengthen our algorithm's approximation ratio for the special case $F = C$, but the general case does not require $F = C$. We also use $d(x, S)$ to denote $\min_{y \in S} d(x, y)$, $[n]$ to denote the set $\{1, 2, \ldots, n\}$, and lastly $[0, 1]$ to denote the set $\{x \in \mathbb{Q} | 0 \le x \le 1\}$.

The idea of fair $k-$center clustering originates from the definition of the classical $k-$center clustering.

**Problem 1.** *The classical $k-$center problem asks to find a set of $k$ centers $S \subset F, |S| = k$ and a mapping $\phi : C \to S$ such as to minimize the following objective function*

$$\min_{S \subset F, |S| = k, \Phi} \quad \max_{v \in C} d(v, \phi(v)).$$

In the classical $k-$center problem, it is trivial to show that for a set of centers $S$, the function $\phi$ mapping a point $v \in C$ to its nearest neighbor in $S$ minimizes the $k-$center objective.

The fair $k-$center requires that no group is over-represented or under-represented within any cluster in addition to the classical $k-$center requirement, its definition is as following:

**Problem 2.** *Definition 1 in [5]. In the fair $k-$center clustering problem, we are given a set of $l$ protected groups, $C_1, \ldots, C_l$ (not necessarily disjoint). Each point $c \in C$ belongs to at least one*

group $C_i$. We are also given a set of vectors $\alpha, \beta \in \mathbb{R}^l, \alpha_i \geq \beta_i$. The problem requires finding a set $S \subset F, |S| = k$, and a mapping $\phi : C \to S$ such as to minimize the following objective function

$$
\begin{aligned}
&\min_{S \subset F, |S| = k, \phi} && \max_{v \in C} d(v, \phi(v)) \\
&\quad s.t && |\{v \in C_i | \phi(v) = f\}| \leq \alpha_i |\{v \in C | \phi(v) = f\}| \quad \forall f \in S, \forall i \in [l] \quad (RD), \\
& && |\{v \in C_i | \phi(v) = f\}| \geq \beta_i |\{v \in C | \phi(v) = f\}| \quad \forall f \in S, \forall i \in [l] \quad (MP),
\end{aligned}
$$

where the RD is the restricted dominance constraint, and MP is the minority protection constraint.

While mapping a point to its nearest center minimizes the cost, it does not guarantee that the resulting clusters would satisfy the RD and MP constraints. Let $\Delta$ denote the maximum number of groups a single client $v \in C$ can belong to. We note that in the Ahmadian et al.[2] paper, they studied the variant with the set of restriction $V = \{\beta = \mathbf{0}, \Delta = 1, \alpha_i = \alpha \forall i \in [l]\}$. Bera et al. [5] noted that Problem 2 is NP-Hard via a reduction from the 3D matching problem, while [2] showed that their restricted problem is also NP hard to approximate with a ratio better than 2 for $\alpha \in (0, 0.5]$. Hence, it is natural to consider a relaxation of the fair $k-$center formulation:

**Problem 3.** *The $\mathcal{E}-$relaxed fair $k-$center clustering problem is the same as Problem 2, but it relaxes the RD and MP constraints to allow an $\mathcal{E}$ additive violation to the fairness constraint*

$$
\begin{aligned}
&\min_{S \subset F, |S| = k, \phi} && \max_{v \in C} d(v, \phi(v)) \\
&\quad s.t && |\{v \in C_i | \phi(v) = f\}| \leq \alpha_i |\{v \in C | \phi(v) = f\}| + \mathcal{E} \quad \forall f \in S, \forall i \in [l] \quad (RD), \\
& && |\{v \in C_i | \phi(v) = f\}| \geq \beta_i |\{v \in C | \phi(v) = f\}| - \mathcal{E} \quad \forall f \in S, \forall i \in [l] \quad (MP),
\end{aligned}
$$

*for some additive violation $\mathcal{E}$.*

Let $\lambda^*, \lambda^*_\alpha, \lambda^*_{\alpha, \mathcal{E}}$ be the optimal $k-$center distance for Problem $1, 2, 3$ respectively for the same input set $(C, F, k)$. Then it is clear that

$$\lambda^* \leq \lambda^*_{\alpha, \mathcal{E}} \leq \lambda^*_\alpha.$$

Ahmadian et al. [2] provided a $3-$approximation algorithm using linear programming and rounding for Problem 2 under the restrictions $V$, with the caveat that the resulting solution is feasible in Problem 3 with $\mathcal{E} = 2$. In other words, their algorithm admits a clustering that violates the constraints additively by at most 2 points, and has $k-$center cost at most $3\lambda^*_\alpha$. However, the linear program described in [2] has $\Theta(N^2)$ variables and $\Theta(N^2)$ constraints for the linear program which make it not scalable for large inputs. They also provided an alternative practical variation of their algorithm that has $\Theta(Nk)$ variables and constraints, but the $3-$approximation analysis does not hold for this variation.

On the other hand, Bera et al. [5] provided a general framework to solve Problem 2 for multiple clustering costs ($k - \{$center, means, median, $...\}$), which corresponds to a $5-$approximation to Problem 2 with the caveat that the additive violation is at most $4\Delta + 3$. Their approximation ratio improves to 4 if $F = C$.

Lastly, Bercea et al. [6] provided a $3-$approximation algorithm when $F = C$ and $5-$approximation when $F \neq C$ for Problem 2. However, the LP uses $\Theta(N^2)$ LP variables and constraints, and does not allow a point to belong to multiple groups, i.e $\Delta = 1$. While the algorithm is elegant, it involves a mixture of advanced and computation heavy techniques such as a large LP, and then using a max-flow min cost setup to round the LP variables which makes it impractical for large input sizes.

In this paper, we present a new scalable randomized $5-$approximation algorithm to Problem 2, which improves to a $3-$approximation if $F = C$. Although the clustering returns by our algorithm might also not be feasible for Problem 2, we prove that it satisfies $\mathbb{E}(\mathcal{E}) = 0$ so it should satisfy the constraints on average. We also sketch how to alter our algorithm to enforce a maximum additive violation of $4\Delta + 3$, as opposed to just $\mathbb{E}(\mathcal{E}) = 0$, using the LP iterative rounding technique developed by Bera. et al. [5]. We show experimentally on real datasets that our algorithm beats the state of the art algorithms in terms of runtime (by orders of magnitude) and clustering cost, while achieving very similar additive violations $\mathcal{E}$ close to 0 in the resulting clusters.

## 3 Definitions and Algorithm

Our algorithm for Problem 2 is outlined in Algorithm 1. In essence, we first run the greedy $k-$center algorithm for Problem 1 to get $k$ centers $S \subset F$. Using the fact that $\lambda^*_\alpha \in [0, 2 \max_{x,y}(d(x, y))]$, we

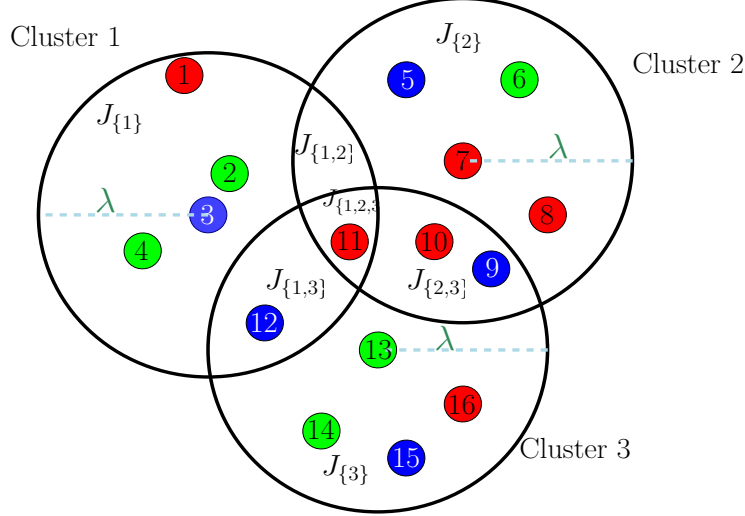

Figure 1: A $\lambda-$Venn diagram for the three centers $\{3, 7, 13\}$, where point 6 *belongs* to $J_{\{2\}}$, point 12 *belongs* to $J_{\{1,3\}}$, point 11 *belongs* to $J_{\{1,2,3\}}$. In total, the joiners are $J_{\{1\}}, J_{\{2\}}, J_{\{3\}}, J_{\{1,2\}}, J_{\{1,3\}}, J_{\{2,3\}}, J_{\{1,2,3\}}$. All joiners are not empty except $J_{\{1,2\}}$

do binary search on the optimal radius $\lambda_\alpha^*$. In each step, we consider a guess for the optimal radius $\lambda$, and formulate a **frequency-distributor** linear program (to be explained later) of a small size. Then we update the binary search interval based on whether the LP has a feasible solution or not. In what follows, we fix a $\lambda$ and assume $d(x, S) \leq \lambda, \forall x \in C$. For the ease of our analysis, let us define a $\lambda-$Venn diagram:

**Definition 1.** *Given a set of $k$ centers $S$, a $\lambda-$Venn diagram region is the region $R$ defined as*

$$R \triangleq \bigcup_{x \in S} B(x, \lambda),$$

*where $B(x, \lambda)$ is the closed ball with center $x$ and radius $\lambda$.*

Now fix a $\lambda-$Venn diagram that includes all the $N$ points, we define a **joiner** object as follows:

**Definition 2.** *Given a nonempty subset $S' \subset S$, denote $J_{S',\lambda}$ as the **joiner** of clusters $S'$ as the region*

$$J_{S',\lambda} \triangleq \bigcap_{x \in S'} B(x, \lambda) \bigcap_{x \in S-S'} \overline{B(x, \lambda)},$$

*where $\overline{B(x, \lambda)}$ is the compliment of the ball.*

We say a point $x$ belongs to a joiner $J_{S,\lambda}$ if $x \in J_{S,\lambda}$ and that a joiner is empty if no point from $C$ belongs to it. Figure 1 shows a diagram illustrating these definitions.

Next, we state the following topology lemma. (See Appendix $A$ for proofs.)

**Lemma 1.** *Given a set of $k-$centers $S$, and an associated $\lambda-$Venn diagram $R$, denote $2^S$ as the powerset of $S$, then the following holds*

1. *For non-empty $A, A' \subset S, A \neq A'$, we must have $J_{A,\lambda} \cap J_{A',\lambda} = \phi$*

2. *The union of all joiners partitions $R$: $R = \bigcup_{A \in 2^S, A \neq \phi} J_{A,\lambda}$*

3. *The number of non empty joiners is at most $\min(2^k - 1, N)$*

Intuitively, the concept of a joiner is useful because for a point $x_i \in C$, and fixed $\lambda$, $x_i \in J_{A,\lambda}$ if and only if $x_i$ can be assigned to a cluster center $f \in A$ and cannot be assigned to a cluster center $f \in S - A$, with a radius $\leq \lambda$.

Next, we define the signature of a point $x \in C$ as a tuple of indices (ordered in increasing order) of all the groups it belongs to:

**Definition 3.** *For any point $x \in C$ that belongs to the groups $C_{i_1}, \ldots, C_{i_r}$ with $i_1 \leq \ldots \leq i_r$, then the signature of $x$ is defined as $Sig(x) = (i_1, \ldots, i_r)$*

By definition, we must have $r \leq \Delta$ for any point. We also use $I$ to denote all possible signatures, with $|I| \leq N$ as each point only has one signature. We say a group $C_i$ belongs to signature $c$ if $i$ is inside the signature ($i \in c$)

Now for any fixed signature $c \in I$ and $S' \subset S$, consider the set of points

$$L(c, S', \lambda) = \{x_i \in C | x_i \in J_{S', \lambda} \text{ and } Sig(x_i) = c\}.$$

Given any solution to Problem 2 with radius $\lambda$, the points in $L(c, S', \lambda)$ are *interchangeable*. This means that a solution to Problem 2 assigning $a_j$ points of $L(c, S', \lambda)$ to cluster $j \in S'$ is a valid solution only if assigning any $a_j$ points of $L(c, S', \lambda)$ to cluster $j$ would also be a valid solution.

The next question to answer is, what portion of the $|L(c, S', \lambda)|$ points should be assigned to which cluster center $f \in S'$. To answer this, we define a **frequency-distributor** linear program.

---

**Algorithm 1** Fair $k$ clustering

1: **function** FAIR_K_CLUSTER($C, F, k, \alpha, \beta, \epsilon$)
2:     $S \leftarrow greedy\_k\_center(F, k)$
3:     $d \leftarrow distance\_matrix(C, S)$
4:     $l, r, feasible \leftarrow 0, 2 \max(d), False$
5:     **while** $r - l > \epsilon$ or not $feasible$ **do**
6:         $\lambda \leftarrow \frac{l+r}{2}$
7:         **if** $\exists x \in C$ with $d(x, S) > \lambda$ **then**
8:             $l, feasible \leftarrow \lambda, False$
9:             continue
10:        $LP \leftarrow Frequency\_Distributor\_LP(C, S, k, \alpha, \beta, \lambda)$
11:        **if** $LP$ has a feasible solution **then**
12:            $r, feasible \leftarrow \lambda, True$
13:        **else**
14:            $l, feasible \leftarrow \lambda, False$
    **return** $\lambda$

---

**Definition 4.** *For each signature $c \in I$, non-empty $S' \subset S$, and $j \in S'$ such that $|L(c, S', \lambda)| > 0$, define an LP variable $x_{c,S',j}$ denoting the number of points in $J_{S',\lambda}$ with signature $c$ that would be assigned to cluster $j \in S'$. Define the **frequency-distributor** linear program as follows:*

$$\min \quad 1$$

$$\beta_a \sum_{\substack{c \in I, S' \in 2^S \\ |j \in S'}} x_{c,S',j} \leq \sum_{\substack{S' \in 2^S, c \in I \\ |j \in S', a \in c}} x_{c,S',j} \leq \alpha_a \sum_{\substack{c \in I, S' \in 2^S \\ |j \in S'}} x_{c,S',j} \qquad \begin{matrix} \forall a \in [l] \\ \forall j \in S \end{matrix} \qquad (1)$$

$$\sum_{j \in S'} x_{c,S',j} = |L(c, S', \lambda)| \qquad \begin{matrix} \forall c \in I, \forall S' \in 2^S \\ \text{such that } |L(c,S',\lambda)| > 0 \end{matrix} (2)$$

$$x_{c,S',j} \geq 0 \qquad \begin{matrix} \forall c \in I, S' \in 2^S, \forall j \in S' \\ \text{such that } |L(c,S',\lambda)| > 0 \end{matrix} (3)$$

The set of constraints $(1)$ defines the fairness constraint on the assignment. The set of constraints $(2)$ ensures that all points get assigned to a cluster. To get an idea on how large the frequency-distributor linear program is, we prove the following lemmas to bound the number of variables and constraints:

**Lemma 2.** *The number of variables is at most $\min(2^{k-1}k|I|, Nk)$*

*Proof.* See Appendix $B$.                                                                                          □

Now, we bound the number of constraints.

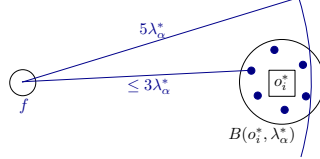

Figure 2: Visual idea of Theorem 1 proof

**Lemma 3.** *The number of constraints is at most* $kl + \min(2^k|I|, Nk) + \min(2^{k-1}k|I|, Nk)$

*Proof.* See Appendix $B$. □

Note that compared to the $LPs$ described in [2, 5, 6], our LP uses at most $\min(2^{k-1}k|I|, Nk)$ variables, as opposed to $\Theta(N^2), \Theta(Nk), \Theta(N^2)$ respectively; and at most $kl + \min(2^k|I|, Nk) + \min(2^{k-1}k|I|, Nk)$ constraints, as opposed to $\Theta(N^2), \Theta(Nk)$, and $\Theta(N^2)$ respectively.
In practice, the frequency-distributor linear program tends to be very small, both in number of variables and constraints, which leads to a quick feasibility check in Algorithm 1.

### 3.1 Frequency-distributor example for $k = 3$

See Appendix $C$ for a step-by-step example of formulating a frequency-distributor LP for the example in Figure 1. We omit it from the main paper due to the page-limit.

### 3.2 Randomized Assignment

Suppose the frequency-distributor LP has a feasible solution **x**. For every non-empty $S' \subset S$, center $j \in S'$, signature $c \in I$, point $x_i \in L(c, S', \lambda)$, we **independently** assign $x_i$ to cluster $j$ with probability $\frac{x_{c,S',j}}{|L(c,S',\lambda)|}$. Thus, on expectation, cluster $j$ would be assigned to $\frac{x_{c,S',j}}{|L(c,S',\lambda)|} \times |L(c, S', \lambda)| = x_{c,S',j}$ points, for all $c, S'$, leading to each cluster $j$ receives $x_{c,S',j}$ points with signature $c$ from each joiner region $J_{S'}$ on expectation as well. Since $x_{c,S',j}$ is a feasible solution that respects the fairness constraints, the randomized assignment respects the RD and MP constraints on expectation using linearity of expectation for each cluster $j$.

Note that we can also restrict the number of violations to $4\Delta + 3$, as opposed to just $\mathbb{E}(\mathcal{E}) = 0$, using iterative rounding in a similar fashion to Bera. et al. [5]. The proof is deferred to the extended paper.

## 4 Analysis

In this section, we prove our algorithm is a $5-$approximation to Problem 2, and a 3-approximation for the case $F = C$. We defer the runtime analysis to Appendix $D$ due to the page limit.

### 4.1 $5-$Approximation

**Theorem 1.** *Algorithm* 1 *is a* $5-$*approximation to Problem* 2*, and a* $3-$*approximation when* $F = C$.

*Proof.* Let $O^* = \{o_1, o_2, ..., o_k\}$ be the optimal centers for Problem 2, $\lambda_\alpha^*$ be the optimal radius, and $\phi^*$ be the optimal mapping. This partitions $C$ into $k$ partitions $P_1, .., P_k$ where $P_i = \{v \in C | \phi^*(v) = o_i\}$. Consider the $k$ clients chosen by the greedy algorithm as $U$, and the corresponding $k$ centers as $S = \{\sigma(u) | u \in U\}$, where $\sigma : C \to F$ denotes the closest center from $F$ to $u$.

Consider an optimal cluster $B(o_i^*, \lambda_\alpha^*)$ and the greedy centers $S$. To prove the LP admits a solution, it is sufficient to prove that there is a greedy center $f \in S$ such that $B(o_i^*, \lambda_\alpha^*) \cap C \subset B(f, 5\lambda_\alpha^*)$ as illustrated in figure 2). This together with Claim 4 from Bera et al. [5] gurantees the existence of a solution.

If for every optimal center $o_i$, there is one client $u_i \in U$ in the ball $B(o_i, \lambda_\alpha^*)$, then the ball $B(\sigma(u), 3\lambda_\alpha^*)$ contains the entirety of $B(o_i, \lambda_\alpha^*)$, because for any point $x \in B(o_i, \lambda_\alpha^*), d(\sigma(u), x) \le d(\sigma(u), u) + d(u, o_i) + d(o_i, x) \le 3\lambda_\alpha^*$. Thus, the frequency distributor LP admits a feasible

assignment which all the points in the optimal partition $P_i$ to the greedy cluster are contained in. This is possible because for every point $x \in P_i$ belonging to joiner $J_{S',\lambda}$ with $sig(x) = c$, then the point can contribute 1 to $x_{c,S',i}$. Note that if $F = C$, the optimal ball $B(o_i, \lambda_\alpha^*)$ is contained in the ball $B(u_i, 2\lambda_\alpha^*)$, because $\sigma(u) = u$.

On the other hand, consider if one optimal cluster has more than one client $u_a, u_b \in U$. Assume this first happens in optimal cluster with center $o_i$ and WLOG assume $a \leq b$. We have that $d(\sigma(u_a), u_b) \leq d(\sigma(u_a), u_a) + d(u_a, o_i) + d(o_i, u_b) \leq 3\lambda_\alpha^*$. But the greedy algorithm chooses $u_b$ because it is the furthest point from any of the chosen centers so far, meaning that every client left is now within $3\lambda_\alpha^*$ from some greedy center in $S$. Let $L = \{i | \forall u \in U, u \notin B(o_i, \lambda_\alpha^*)\}$ be the clusters left without a client from $U$ in them. Let $\tau : L \to C$ be any function that arbitrarily chooses any client inside the remaining clusters $L$. Then the balls $B(\sigma(\tau(i)), 5\lambda_\alpha^*)$ completely encompasses the remaining clusters $W = \{B(o_i, \lambda_\alpha^*) | i \in L\}$. Note that if $F = C$ we can improve the bound. First, all the remaining points are reachable from $u_a$ with distance at most $2\lambda_\alpha^*$ from the first paragraph. Thus, all centers (which are also clients) $\{o_i | i \in L\}$ are at a distance at most $2\lambda_\alpha^*$ from $\sigma(u_a)$. So the ball $B(u_a, 3\lambda_\alpha^*)$ completely encompasses the remaining clusters. Since the clusters are completely enclosed in some greedy cluster, the frequency distributor LP admits a valid solution at $\lambda = 5\lambda_\alpha^*$, or $\lambda = 3\lambda_\alpha^*$ when $F = C$ using the assignment from Claim 4 from Bera et al. [5]. $\square$

## 5 Empirical Evaluation

### 5.1 Experimental Setup

We conduct experiments on 6 real-world datasets: reuters, victorian, 4area from [2], and bank, census, creditcard from [5].

| Dataset | reuters | victorian | 4area | bank | census | creditcard |
|---|---|---|---|---|---|---|
| Sample size | 2,500 | 4,500 | 35,385 | 4,512 | 32,561 | 30,000 |
| Features (dimension) | 10 | 10 | 8 | 3 | 5 | 13 |
| Number of Groups | 50 | 45 | 4 | 5 | 7 | 8 |
| Overlapping groups | No | No | No | Yes | Yes | Yes |
| Protected group feature(s) | author | author | author | marital, default | race,sex | marriage, education |

Table 1: Statistics of baseline datasets

**Baseline Algorithms**. We use three baseline algorithms in our experiment. The first one is the greedy $k-$center algorithm, note that it does not respect the fairness constraint, and so its cost is likely to be less than a cost that respects the constraint. The second one is the practical Linear Program from [2] (Not the $3-$ approximation). Note that we did not include the algorithm from [6] as it TLE on almost all input instances as was the case for the $3-$approximation from [2]. This is mainly due to them using $\Theta(N^2)$ LP variables. Finally, we include the $5-$approximation linear program from [5].

**Metrics**. We evaluate our algorithm and the baselines based on 3 metrics. ***Cost***. Defined as the $k-$center cost, or the maximum distance from a point to its nearest center reported by the algorithm. ***Runtime***. We report the runtime in seconds that the algorithm take to terminate. If any algorithm takes longer than 30 minutes per run, we report a Time limit exceeded (TLE). ***Additive Violation of fairness constraint***. We report the maximum additive violation $\mathcal{E}$ in any cluster output by all algorithms. Recall that for our algorithm, $\mathbb{E}(\mathcal{E}) = 0$, while the LP from [2], $\mathcal{E} \leq 2$ and the LP from [5], $\mathcal{E} \leq 4\Delta + 3$.

#### 5.1.1 Implementation Details

The implementation for our algorithm and Ahmadian et al. [2] is available at [14]. As the original implementation of [2] was not available publicly or upon request, we implement [2] based on their paper. We use the implementation of Bera et al. [5] available at [23] for the $5-$approximation. All algorithms are written in Python 3. For our algorithm and [2], we use the CPLEX solver to handle the linear programs in our implementation which drastically improves the solution speed, with the linear programs defined using the library Pulp [22], while the original implementation from [5] also uses CPLEX. We modify it to accept $\alpha, \beta$ as input parameters. All the computations are run independently on a Macbook Pro with a 2.4 GHz Quad-Core Intel Core i5 processor.

## 5.2 Experimental Results

| Dataset | $\alpha$ | Cost | | | | $\mathcal{E}$ | | | | Runtime(s) | | | |
|---|---|---|---|---|---|---|---|---|---|---|---|---|---|
| | | KFC | G | A | B | KFC | G | A | B | KFC | G | A | B |
| reuters | 0.05 | **1.858** | 1.593 | 3.679 | 1.865 | 1(2) | 18 | **0**(0) | 2(2) | 77.661 | 0.173 | **51.54** | 61.77 |
| | 0.20 | **1.573** | 1.575 | 3.278 | 1.604 | 1(1) | 10 | **0**(0) | **0**(0) | **30.085** | 0.172 | 44.68 | 53.56 |
| | 0.40 | **1.540** | 1.582 | 3.278 | 1.604 | 1(1) | 3 | **0**(0) | 1(1) | **34.922** | 0.174 | 47.19 | 53.26 |
| victorian | 0.1 | 4.698 | 3.240 | 6.958 | **4.602** | 2(2) | 35 | **0**(0) | 1(1) | 158.2 | 0.314 | **89.65** | 180.3 |
| | 0.3 | **3.868** | 3.108 | 7.612 | 4.174 | 1(2) | 21 | **0**(0) | 1(1) | **81.53** | 0.309 | 88.87 | 165.6 |
| | 0.5 | **3.580** | 3.228 | 6.958 | 3.820 | 1(1) | 8 | **0**(0) | **0**(0) | **76.88** | 0.308 | 81.16 | 142.3 |
| 4area | 0.45 | **9.421** | 9.786 | 19.13 | 9.768 | 2(2) | 31 | **0**(0) | **0**(0) | **14.51** | 2.283 | 393.1 | 1190 |
| | 0.60 | 9.891 | 9.535 | 18.90 | **9.768** | **0**(1) | 0 | **0**(0) | **0**(0) | **12.96** | 2.237 | 370.4 | 1276 |
| | 0.80 | **9.671** | 9.762 | 14.02 | 9.768 | **0**(1) | 0 | **0**(0) | **0**(0) | **12.06** | 2.224 | 362.4 | 1205 |
| bank ,cost x10$^4$ | 0.80 | **2.914** | 0.133 | N/A | 7.450 | **1**(1) | 1 | N/A | 504(504) | **2.941** | 0.292 | N/A | 117.5 |
| | 0.90 | **2.914** | 0.133 | N/A | 7.450 | **1**(1) | 1 | N/A | 231(231) | **2.940** | 0.291 | N/A | 117.9 |
| | 1.00 | **0.121** | 0.127 | N/A | N/A | **0**(0) | 0 | N/A | N/A | **2.496** | 0.289 | N/A | N/A |
| census ,cost x10$^4$ | 0.86 | **118.5** | 58.34 | N/A | TLE | **0**(0) | 129 | N/A | TLE | **19.83** | 2.153 | N/A | TLE |
| | 0.90 | **118.5** | 57.33 | N/A | TLE | **0**(0) | 1 | N/A | TLE | **19.63** | 2.168 | N/A | TLE |
| | 0.94 | **118.5** | 59.87 | N/A | TLE | **1**(1) | 1 | N/A | TLE | **20.11** | 2.144 | N/A | TLE |
| creditcard ,cost x10$^4$ | 0.60 | **124.5** | 56.02 | N/A | TLE | **1**(5) | 15 | N/A | TLE | **26.49** | 2.472 | N/A | TLE |
| | 0.70 | **124.5** | 58.22 | N/A | TLE | **1**(6) | 1 | N/A | TLE | **26.78** | 2.456 | N/A | TLE |
| | 0.80 | **124.5** | 56.85 | N/A | TLE | **1**(8) | 1 | N/A | TLE | **26.83** | 2.485 | N/A | TLE |

Table 2: Comparison of $\lambda$, $\mathcal{E}$, and runtime(sec) of our algorithm with 3 baselines with varying $\alpha$

The first experiment is a comparison with the baselines with respect to the fairness constraint $\alpha$, where the parameters are fixed as $k = 25, \epsilon = 0.1, \beta = 0$, and $\alpha$'s with feasible solutions are selected. Note that we abbreviate greedy as G, Ahmadian et al. algorithm [2] by A, and Bera et al.[5] algorithm by B, N/A as no solution found, and TLE as time limit exceeded for limit equals to 30 minutes. In addition, as A does not handle overlapping groups, it is not run on the last three datasets. For each dataset and $\alpha$, we run each experiment 5 times and report the average cost, runtime, and median $\mathcal{E}$, where the maximum $\mathcal{E}$ from the 5 runs is enclosed in brackets to show the robustness of our algorithm.

As shown in Table 2, our algorithm, A and B achieve a cost close to greedy and $\mathcal{E}$ within 2. Our result is also in sync with the results in [2] and [5] that the $\mathcal{E}$ of greedy can be very high for tight $\alpha$-cap. Our algorithm also runs significantly faster on datasets with small number of groups shown by 4area.

The second experiment is the effect of $k$ on the cost and runtime. We use $\alpha = 0.5, \epsilon = 0.1$ on reuters and victorian with varying $k$. Aligned with the experimental results in [2], the runtime increases with $k$ while the cost decreases as $k$ increases. As shown in figure 3, our algorithm and B follow the cost of greedy tightly until they reach a limit as k grows large, whereas A incurs a cost in double. In terms of runtime, A is the fastest followed by ours and then B. Overall out algorithm gives the best balance of run-time speed and cost guarantees.

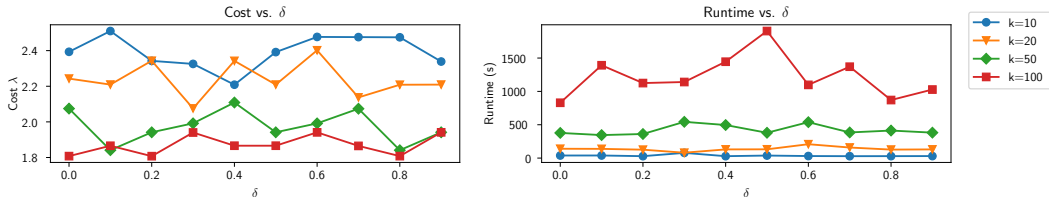

Figure 4: Cost and runtime of our algorithm over varying $\delta$ for $\epsilon = 0.1$ on reuters

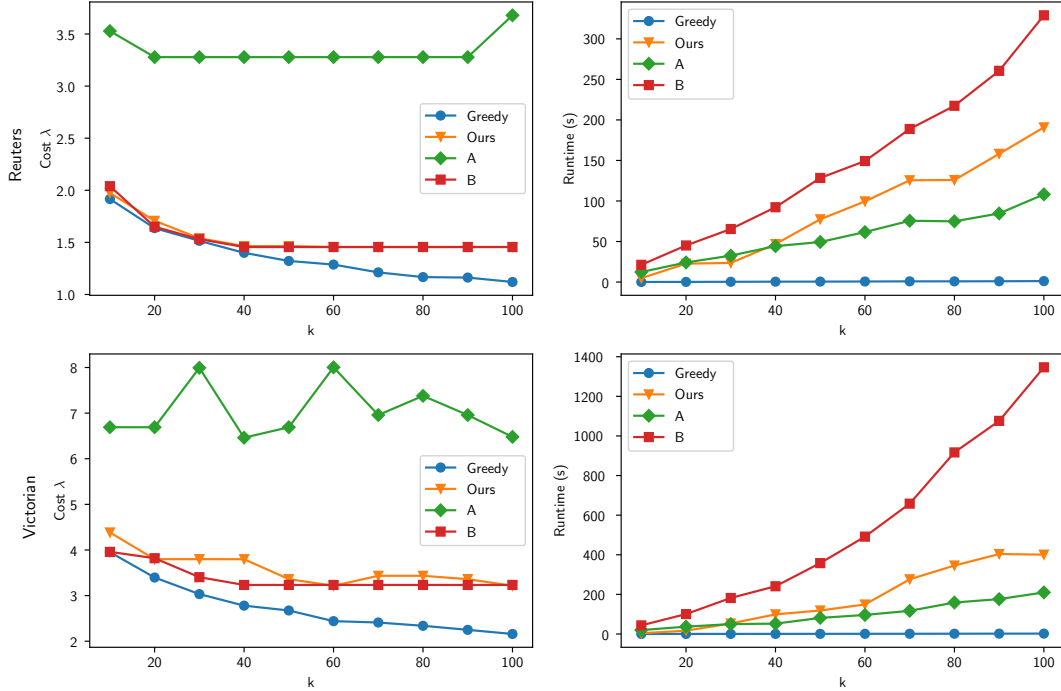

Figure 3: Cost and runtime of the solution vs. three baselines on reuters and victorian for varying $k$, using $\epsilon = 0.1, \alpha = 0.5, \beta = 0$

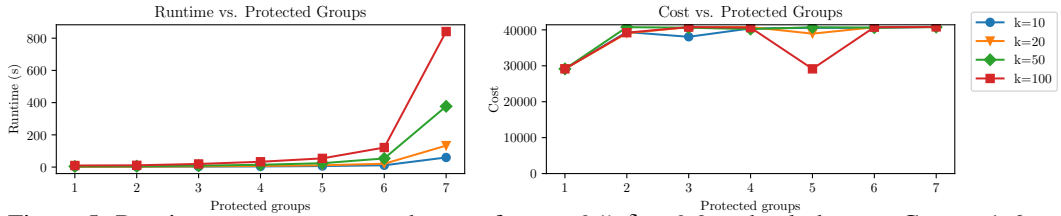

Figure 5: Runtime, cost vs. protected group for $\epsilon = 0.5, \delta = 0.2$ on bank dataset. Groups 1-6 are either binary (2 categories) or tertiary (3 categories). Group 7 has 11 categories.

The third result is the performance of our algorithm with changing $\delta$ value, where $\delta$ is a parameter defined by Bera et al. to control $\alpha_i$ and $\beta_i$, with $\beta_i = r_i(1 - \delta)$ and $\alpha_i = r_i/(1 - \delta)$, and $r_i$ as the ratio of size of group $i$ to the total number of points. Figure 4 shows that there is no trend of worsening runtime with respect to tighter $\delta$.

Finally, the effect of the number of protected groups on runtime and cost is shown in figure 5. As expected, the runtime increases with the number of protected groups since the number of unique joiner-signatures pairs increases. At the same time, the costs remain similar possibly due to correlated protected groups leading to unchanging clusters even if the number of protected groups increases.

## 6 Conclusion

In this paper, we present a $k$-{center, supplier} fair clustering algorithm that imposes fairness constraints to ensure no group becomes a majority or minority in any cluster. Our algorithm improves the approximation ratio from 4 to 3, beats the state of the art by several orders of magnitude on several datasets, scales well for large $N$, and respects the fairness constraints on expectation. The LP size used by the algorithm does not necessarily depend on the number of points in the input, but instead on the number of unique joiner-signature pairs in the input.

## 7 Broader Impact

Any clustering algorithm that doesn't take into consideration the underlying bias in data for some minority groups risks producing biased results against these groups. For example, In the United States there are computer programs that predict whether a criminal is likely to re-offend, and is used by judges to decide on the sentence length. One such program is the Correctional Offender Management Profiling for Alternative Sanctions (COMPAS) sold by NORTHPOINTE and was used by judges in Wisconsin. In a report by Pro-Publica, it was shown that COMPAS is racially biased, where it was twice as likely to falsely flag African American defendants as future criminals as much as White defendants, holding everything else constant. This was because the features (137 features representing answers of questions) that were used when clustering potential re-offenders were heavily biased against African American defendants. Several other examples show the necessity of having a fair clustering algorithm that protects minority groups, as well as prevents a certain group from dominating any cluster. This justifies the necessity of studying this problem and proposing new fair algorithms.

Our algorithm puts an effective boundary ensure all groups are neither dominating nor underrepresented, as guaranteed by the $\alpha, \beta$ parameters. It could be useful when we want to maintain diversity in clusters, as pointed out in the marketing and committee selection examples by [2]. However, putting our algorithm in the wrong hands can lead to intensifying the issue. A malicious adversary can restrict certain groups (by changing the $\alpha, \beta$ parameter) to bias the model against having clusters with a representative percentage of a certain group, and thus care must be taken when setting the $\alpha, \beta$ parameters to not pass one's own bias into the clustering algorithm.

## Footnotes

*Both authors contributed equally to the paper. Author order is in alphabetical order. We thank Ho Chung Leon Law from University of Oxford for recommending the algorithm name. We also thank all the reviewers for their incisive comments and helping us improve the paper.

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
