[Supplementary Material · Appendix_Neurips.pdf]

# Appendix

## 1 Cleaner view of Table 2 (Due to Formatting Issues in Main Paper)

| Dataset | $\alpha$ | cost | | | | $\mathcal{E}$ | | | | runtime(s) | | | |
|---|---|---|---|---|---|---|---|---|---|---|---|---|---|
| | | us | greedy | A | B | us | greedy | A | B | us | greedy | A | B |
| reuters | 0.05 | **1.858** | 1.593 | 3.679 | 1.865 | 1 | 18 | **0** | 2 | 77.661 | 0.173 | **51.54** | 61.77 |
| | 0.20 | **1.573** | 1.575 | 3.278 | 1.604 | 1 | 10 | **0** | **0** | **30.085** | 0.172 | 44.68 | 53.56 |
| | 0.40 | **1.540** | 1.582 | 3.278 | 1.604 | 1 | 3 | **0** | 1 | **34.922** | 0.174 | 47.19 | 53.26 |
| victorian | 0.1 | 4.698 | 3.240 | 6.958 | **4.602** | 2 | 35 | **0** | 1 | 158.2 | 0.314 | **89.65** | 180.3 |
| | 0.3 | **3.868** | 3.108 | 7.612 | 4.174 | 1 | 21 | **0** | 1 | **81.53** | 0.309 | 88.87 | 165.6 |
| | 0.5 | **3.580** | 3.228 | 6.958 | 3.820 | 1 | 8 | **0** | **0** | **76.88** | 0.308 | 81.16 | 142.3 |
| 4area | 0.45 | **9.421** | 9.786 | 19.13 | 9.768 | 2 | 31 | **0** | **0** | **14.51** | 2.283 | 393.1 | 1190 |
| | 0.60 | 9.891 | 9.535 | 18.90 | **9.768** | **0** | 0 | **0** | **0** | **12.96** | 2.237 | 370.4 | 1276 |
| | 0.80 | **9.671** | 9.762 | 14.02 | 9.768 | **0** | 0 | **0** | **0** | **12.06** | 2.224 | 362.4 | 1205 |
| bank | 0.80 | **2.914** | 0.133 | N/A | 7.450 | 1 | 1 | N/A | 504 | **2.941** | 0.292 | N/A | 117.5 |
| ,cost x10$^4$ | 0.90 | **2.914** | 0.133 | N/A | 7.450 | **1** | 1 | N/A | 231 | **2.940** | 0.291 | N/A | 117.9 |
| | 1.00 | **0.121** | 0.127 | N/A | N/A | **0** | 0 | N/A | N/A | **2.496** | 0.289 | N/A | N/A |
| census | 0.86 | **118.5** | 58.34 | N/A | TLE | **0** | 129 | N/A | TLE | **19.83** | 2.153 | N/A | TLE |
| ,cost x10$^4$ | 0.90 | **118.5** | 57.33 | N/A | TLE | **0** | 1 | N/A | TLE | **19.63** | 2.168 | N/A | TLE |
| | 0.94 | **118.5** | 59.87 | N/A | TLE | 1 | 1 | N/A | TLE | **20.11** | 2.144 | N/A | TLE |
| creditcard | 0.60 | **124.5** | 56.02 | N/A | TLE | 1 | 15 | N/A | TLE | **26.49** | 2.472 | N/A | TLE |
| ,cost x10$^4$ | 0.70 | **124.5** | 58.22 | N/A | TLE | 1 | 1 | N/A | TLE | **26.78** | 2.456 | N/A | TLE |
| | 0.80 | **124.5** | 56.85 | N/A | TLE | 1 | 1 | N/A | TLE | **26.83** | 2.485 | N/A | TLE |

Note than in the table above. We bold the winner of cost, $\mathcal{E}$, and runtime between our algorithm, as well as Ahmadian et al and Bera et al algorithm. The Greedy algorithm is not included as it is not a fair algorithm as seen from its $\mathcal{E}$, but is left for comparison purposes.

## Appendix $A$

**Lemma 1.** *Given a set of $k-$centers $S$, and an associated $\lambda- Venn$ diagram $R$. Denote $2^S$ as the powerset of $S$, then the following holds*

1. *For non-empty $A, A' \subset S, A \neq A'$, we must have $J_A \cap J_{A'} = \phi$*

2. *The union of all joiners partitions $R$: $R = \bigcup_{A \in 2^S, A \neq \phi} J_A$*

3. *The number of non empty joiners is at most $\min(2^k - 1, N)$*

*Proof.* 1) Without loss of generality, assume $|A| \leq |A'|$. Since $A' \neq A$, there exists an $i \in A'$ such that $i \notin A$, or equivalently, $i \in S - A$. Now take any point $x \in J_{A'}$. By definition, $x \in B(i, \lambda)$ must be true. However, using the fact that a point in $J_A$ must belong to the region $\overline{B(i, \lambda)}$, it follows that $x$ cannot belong to $J_A$

2) Take any point $x \in R$. Let $A$ be the set of centers $A \subset S$ that satisfy $d(i, x) \leq \lambda$. Then the set of centers in $j \in S - A$ satisfy $d(i, x) > \lambda$. This implies that $x$ belongs to $J_A$ by the definition of $J_A$. Since $A \in 2^S, A \neq \phi$, then $x$ belongs to the union of joiners. Also note using (1) that the joiners are disjoint, so they partition $R$.

3) There are at most $2^k - 1$ non-empty elements in the power set of $S$. However, every point from the $N$ points belongs to one joiner only (using (1)). This means that the number of non empty joiners is bounded by the number of points, $N$. The result follows. $\square$

| Region, Color | Number of Points | Region, Color | Number of Points |
|---|---|---|---|
| $J_{\{1\}}$,Red | 1 | $J_{\{3\}}$,Green | 2 |
| $J_{\{1\}}$,Green | 2 | $J_{\{3\}}$,Blue | 1 |
| $J_{\{1\}}$,Blue | 1 | $J_{\{1,3\}}$,Blue | 1 |
| $J_{\{2\}}$,Red | 2 | $J_{\{2,3\}}$,Red | 1 |
| $J_{\{2\}}$,Green | 1 | $J_{\{2,3\}}$,Blue | 1 |
| $J_{\{2\}}$,Blue | 1 | $J_{\{1,2,3\}}$,Red | 1 |
| $J_{\{3\}}$,Red | 1 | | |

Table 1: Frequency for each Joiner-color pair.

# Appendix $B$

**Lemma 2.** *The number of variables is at most* $\min(2^{k-1}k|I|, Nk)$

*Proof.* For a fixed signature $c \in I$, and a set $A \subset S$ of size $i$, there are at most $i$ LP variables $\{x_{c,A,j}|j \in A\}$. There are $\binom{k}{i}$ sets of size $i$, so there are at most $|I|\binom{k}{i}i$ variables for sets of size $i$. Summing this up over the set sizes, we get an upper bound on the number of variables

$$\sum_{i=1}^{k} |I|\binom{k}{i}i = 2^{k-1}k|I|$$

However, each point in the input belongs to one, and only one set $L(a, A)$ (since a point has 1 signature, and belongs to 1 joiner $J_A$), leading to at most $N$ pairs $(a, A)$ satisfying $|L(a, A)| > 0$. For each pair $(a, A)$, there could be at most $k$ variables $\{x_{a,A,j}|j \in A\}$, which bounds the number of variables by $Nk$. Combining the two bounds yields the required result. $\square$

Now, we bound the number of constraints

**Lemma 3.** *The number of constraints is at most* $kl + \min(2^k|I|, Nk) + \min(2^{k-1}k|I|, Nk)$

*Proof.* For the set of constraints (1), there are $kl$ constraints.
For the set of constraints (2), There are $|I|$ signatures, and at most $2^k$ possible $A$, so at most $2^k|I|$ constraints.
However, each point belongs to one, and only one of $L(a, A)$, leading to at most $N$ pairs $(a, A)$ satisfying $|L(a, A)| > 0$, which bounds the number of constraints by $Nk$.
Finally, from Lemma 2, there are at most $\min(2^{k-1}k|I|, Nk)$ variables, and thus at most $\min(2^{k-1}k|I|, Nk)$ for the set of constraints (3).
Combining the bounds for all sets of constraints, we get the desired result. $\square$

# Appendix $C$

In this section, we show the **frequency-distributor** linear program for the example in Figure 1 in the paper. There are 3 non overlapping groups, Red, Green, and Blue. Thus, the signature of any point is of length 1. We also set $F = C, k = 3, \beta_g = 0, \alpha_g = \alpha$. Suppose that the initial centers returned by the greedy algorithm are the points $S = \{3, 7, 13\}$.

For each point $x_i$, make a list $S' \subset S$ of cluster centers $j$ such that $d(x_i, j) \leq \lambda \iff j \in S'$ In other words, $S'$ are the cluster centers reachable from $x_i$. Associate the point $x_i$ and its color with $S'$. A straightforward implementation with a hash map would take $O(Nk)$ time. Table 1 shows a summary after the $O(Nk)$ computation. Notice that $J_{\{1,2\}}$ is missing from the hash map. This is because no point belongs to $J_{\{1,2\}}$.

Finally, we define the corresponding 18 LP variables to the region color pairs in Table 1 (We abbreviate Red, Green, and Blue with R,G,B respectively):

$$\{x_{R,\{1\},1}, x_{G,\{1\},1}, x_{B,\{1\},1}, x_{R,\{2\},2}, x_{G,\{2\},2}, x_{B,\{2\},2}, x_{R,\{3\},3}, x_{G,\{3\},3}, x_{B,\{3\},3}, x_{B,\{1,3\},1}, x_{B,\{1,3\},3}$$
$$x_{R,\{2,3\},2}, x_{R,\{2,3\},3}, x_{B,\{2,3\},2}, x_{B,\{2,3\},3}, x_{R,\{1,2,3\},1}, x_{R,\{1,2,3\},2}, x_{R,\{1,2,3\},3}\}$$

These define the following **frequency-distributor** LP:

$$\min 1$$

$$x_{R,\{1,2,3\},1} + x_{R,\{1\},1} \le \alpha(x_{R,\{1,2,3\},1} + x_{R,\{1\},1} + x_{G,\{1\},1} + x_{B,\{1\},1} + x_{B,\{1,3\},1})$$
$$x_{G,\{1\},1} \le \alpha(x_{R,\{1,2,3\},1} + x_{R,\{1\},1} + x_{G,\{1\},1} + x_{B,\{1\},1} + x_{B,\{1,3\},1})$$
$$x_{B,\{1\},1} + x_{B,\{1,3\},1} \le \alpha(x_{R,\{1,2,3\},1} + x_{R,\{1\},1} + x_{G,\{1\},1} + x_{B,\{1\},1} + x_{B,\{1,3\},1})$$

$$x_{R,\{2,3\},2} + x_{R,\{1,2,3\},2} + x_{R,\{2\},2} \le \alpha(x_{R,\{2,3\},2} + x_{R,\{1,2,3\},2} + x_{R,\{2\},2} + x_{G,\{2\},2} + x_{B,\{2,3\},2} + x_{B,\{2\},2})$$
$$x_{G,\{2\},2} \le \alpha(x_{R,\{2,3\},2} + x_{R,\{1,2,3\},2} + x_{R,\{2\},2} + x_{G,\{2\},2} + x_{B,\{2,3\},2} + x_{B,\{2\},2})$$
$$x_{B,\{2,3\},2} + x_{B,\{2\},2} \le \alpha(x_{R,\{2,3\},2} + x_{R,\{1,2,3\},2} + x_{R,\{2\},2} + x_{G,\{2\},2} + x_{B,\{2,3\},2} + x_{B,\{2\},2})$$

$$x_{R,\{2,3\},3} + x_{R,\{1,2,3\},3} + x_{R,\{3\},3} \le \alpha(x_{R,\{2,3\},3} + x_{R,\{1,2,3\},3} + x_{R,\{3\},3} + x_{G,\{3\},3} + x_{B,\{1,3\},3} + x_{B,\{2,3\},3} + x_{B,\{3\},3})$$
$$x_{G,\{3\},3} \le \alpha(x_{R,\{2,3\},3} + x_{R,\{1,2,3\},3} + x_{R,\{3\},3} + x_{G,\{3\},3} + x_{B,\{1,3\},3} + x_{B,\{2,3\},3} + x_{B,\{3\},3})$$
$$x_{B,\{1,3\},3} + x_{B,\{2,3\},3} + 1 \le \alpha(x_{R,\{2,3\},3} + x_{R,\{1,2,3\},3} + x_{R,\{3\},3} + x_{G,\{3\},3} + x_{B,\{1,3\},3} + x_{B,\{2,3\},3} + x_{B,\{3\},3})$$

$$x_{B,\{1,3\},1} + x_{B,\{1,3\},3} = 1$$
$$x_{B,\{2,3\},2} + x_{B,\{2,3\},3} = 1$$
$$x_{R,\{2,3\},2} + x_{R,\{2,3\},3} = 1$$
$$x_{R,\{1,2,3\},1} + x_{R,\{1,2,3\},2} + x_{R,\{1,2,3\},3} = 1$$

$$x_{R,\{1\},1} = 1, x_{G,\{1\},1} = 2, x_{B,\{1\},1} = 1$$
$$x_{R,\{2\},2} = 2, x_{G,\{2\},2} = 1, x_{B,\{2\},2} = 1$$
$$x_{R,\{3\},3} = 1, x_{G,\{3\},3} = 1, x_{B,\{3\},3} = 1$$

$$\mathbf{x} \ge \mathbf{0}$$

Once we have solved the linear program, then for each point $x_i \in C$, suppose the point has color $c$, and belongs to $J_A$. Then assign the point to cluster $j \in A$ with probability $\frac{x_{c,A,j}}{|L(c,A)|}$. Then on expectation, each cluster would receive $x_{c,A,j}$ points from $L(c,A)$. This leads to a solution that respects the constraints on expectation. Indeed solving the LP above and doing the randomized assignment results in $\mathcal{E} = 0$ in 1000 random assignments.

# Appendix D: Runtime Analysis

Denote $LP(m,n)$ as the time needed to solve a linear programming problem with $m$ variables and $n$ constraints. In Algorithm 1, The basic implementation of the greedy $k-$center algorithm takes $O(Nkd)$ time, where $d$ is the dimension of the input points. The distance matrix $distance\_matrix(X,S)$ calculation takes $O(Nkd)$ time. In addition, there are $O(\log \frac{\max(d)}{\epsilon})$ binary search iterations. In each iteration, Lines $8-11$ of Algorithm 1 takes $O(Nk)$ time. The frequency-distributor LP can be constructed in $O(Nkl)$ time (basic implementation without any special data structures to build the LP) and solved in $LP(\min(2^{k-1}k|I|, Nk), kl + \min(2^k|I|, Nk) + \min(2^{k-1}k|I|, Nk))$. For small $k, I, l$, the number of variables and constraints is very small which leads to a very fast construction and feasability check for the LP.

Hence, the entire algorithm takes on a worse case scenario $O(Nkd + \log(\frac{\max(d)}{\epsilon})(Nkl + LP(Nk, 3Nk)))$.