[Reviews · NeurIPS 2020]

Review 1

Summary and Contributions: This paper presents a new algorithm for fair k-center which: 1) matches the approximation factor of a state of the art algorithm in the general case and 2) improves the approximation factor of a state of the art algorithm in a restricted case. The algorithm is seeded by running greedy k-center and then formulating a linear program, parametrized by lambda, that softly reassigns points to clusters so that the fairness constraint is satisfied. A binary search is performed on lambda, and after the best value is found, points are randomly assigned to clusters proportional to the corresponding values in the solution to the LP. The authors show favorable experimental results on six datasets in comparison to two similar algorithms.

Strengths: The algorithm proposed matches the current state of the art in terms of approximation factor in the general case, and improves the approximation factor (from 4 to 3) in the specific case. Also, the algorithm can handle points that can be members of more than one group (unlike one of the state of the art algorithms compared). Another strength is that the proposed algorithm performs favorably in empirical experiments: it consistently achieves a small amount of fairness violation and, in all but 2 cases, is the fastest of the fair algorithms (often by a significant margin).

Weaknesses: The proposed algorithm only satisfies the fairness constraints in expectation. This may lead to unfair outcomes and also makes comparisons with algorithms that always satisfy the constraints somewhat unfair. Similarly, I’m concerned that the variance associated with the fairness violation could be very high. While the authors demonstrate that in experiments, their algorithm achieves comparable fairness violations to the other baseline algorithms, I naturally think about whether there exist datasets where the proposed algorithm would return highly unfair solutions. Additionally, the authors only report the median cost of 5 runs and do not report error bars, which detracts from my confidence in the consistency of results. In theory, their algorithm could be run multiple times and the best run selected, but then this must be added to the reported runtimes. The writing can be improved (more below). Many details are missing from the third experiment. EDIT AFTER AUTHOR RESPONSE: thank you for addressing my concerns and providing some additional experimental details. Also, thank you for clarifying bounds with respect to fairness constraints. I still believe that errors bars should be added in a subsequent version of this paper (which you have agreed to add).

Correctness: As far as I can tell, the theoretical claims are correct. In terms of empirical methodology, I would encourage the authors to run more than 5 restarts of their algorithm. This will make their results more robust. Also, they should report error bars since the algorithm is randomized.

Clarity: I am able to follow the ideas in the paper, but the writing can be improved. Figure 1 does not have enough caption text: it does not specifically illustrate joiners, which the text says it does. The example in the appendix really helped me to understand; perhaps try to find a way to include this in the main paper, especially in that figure. Also, in the experiments section you reference a parameter \epsilon = 0.1; to what does this parameter correspond? There is barely any detail about the third experiment (what is \delta?). If included, this must be fleshed out.

Relation to Prior Work: Yes, as far as I can tell, the authors compare against relevant related approaches. The authors also do a reasonable job describing the line of work that led to their proposed algorithm as well as the differences between their algorithm and previous work.

Reproducibility: Yes

Additional Feedback: Even though I checked yes, I have mixed feelings about reproducibility. I think the algorithm is relatively clear, but the Github link to the code is broken and TLE is defined twice, differently. Also, I don't know how many steps of binary search are run. However, I note that the code is provided, which increases my confidence. I’m curious for further details about why your algorithm is so much faster than the other fair algorithms. Does it simply have to do with fewer variables/constraints in the LPs? I find this gap surprising because you do a binary search where at each step you have to solve an LP. The motivations of clustering for recidivism prediction or loan awards is not realistic (even though you cite previous work). As far as I know, these applications are not solved via clustering. Please add a conclusion. I would suggest changing the plots in Figure 2 to be cost vs. runtime with various operating points representing the different k’s. Implementation details can go to the appendix to make room for other important items (noted in the review). I realized that there is a nicer version of Table 2 in the Appendix; I would try hard to get a nice version in the main paper, e.g., abbr TLE as -, shorten dataset names, etc. You say TLE is 1hr in 5.2.1 and then later say that it’s 30 minutes. I’d be interested to see an experiment that compares the algorithms as the number of protected groups increases. Bibliography reference 5 is strange, please clean this up. I like that the authors mention that this is the over/under representation constrained k-center. This helps mitigate issues stemming from a practitioner relying on any output from the algorithm being “fair” in an absolute sense.


Review 2

Summary and Contributions: In short, I think it's fair to say that this is a more practical variant of the Bera et al. algorithm for fair clustering; benefiting from a more compact fair-assignment LP and direct support for overlapping classes. The paper is concerned with finding approximate clusterings for a scenario where all data points are annotated with classes (topics, keywords, opinions, minorities,...) and we look for a clustering that treats all classes "fairly". There are different notions of fairness around. This paper works along the lines of the Chierichetti et al./Bercea et al./Bera et al. model and asks for fair representation: For each class i, the problem input contains a lower bound alpha_i and an upper bound beta_i and asks for a clustering such that in each cluster c and for each class i, the fraction of points of class i in cluster c is at least alpha_i and at most beta_i. This makes sure that each class is neither under-represented nor over-represented in any cluster. The authors analyze the k-center objective, i.e. they look for a clustering that is fair in the above sense and strive to minimize the largest radius of any cluster. The paper provides a 3-approximation algorithm for the case that any point may be used as a center; if the center set and the point set are distinct (k-supplier), the algorithm is a 5-approximation at worst. In both cases, the algorithm may suffer from a small additive fairness violation. The algorithm works along similar lines as previous works: It starts from an approximative solution X that ignores the fairness constraints and then (repeatedly) solves a linear program to re-assign the points in such a way that the fairness constraints are respected. In that way, the algorithm attains a solution that opens the same centers as X, but is fair in the above sense. Computational experiments at the end of the paper confirm that the presented algorithm produces solutions that are only slightly more expensive to the Bera et al. algorithm (surprising, given that the theoretical approximation guarantee of the Bera et al. algorithm is worse), while being significantly faster. The Bercea et al. algorithm is not included in the experiments.

Strengths: In terms of approximation guarantee, the algorithm is an improvement over a previous 4-approximation by Bera et al. (NeurIPS 2019); however, the authors fail to mention that there is another 3-approximation/5-approximation by Bercea et al. (APPROX 2018). The improvement here is that the present algorithm allows point classes to overlap (to cite the authors, a point may belong to both the "male" and the "asian" class) whereas the Bercea et al. algorithm a priori does not; however, overlapping classes can always be made distinct in preprocessing (in the above example, introduce a "male-asian" class) at the expense of some overhead. Both Bera et al. and Bercea et al. exhibit a small additive fairness violation as well. The remaining (perhaps more important) improvement over the previous works is that the fair assignment LP may be more compact due to the following observation: Suppose an optimum fair clustering has radius lambda. Then, for two points x,y, let S_x and S_y be the set of centers of S that lie within radius lambda of x and y, respectively. If S_x = S_y and x and y belong exactly to the same combination of classes (the authors say that x and y have the same signature), then x and y may be used interchangeably. Following through with this thought yields an LP that aggregates the interchangeable points: Instead of keeping track of where each individual point is assigned, it only ensures that the correct *number* of points of each type is assigned. If we have n points, m class signatures and want k centers, then the compact LP is roughly of size O( min(2^k*k*m, nk) ) whereas the Bercea et al. LP is of size O(n^2). The Bera et al. LP is of size O(nk). Edit after author feedback: I agree with the authors that the proposed preprocessing does not work.

Weaknesses: Both the Bera et al., and the Bercea et al. analysis works for a range of clustering problems (with problem dependent approximation guarantee), whereas the analysis at hand is given only for k-center/k-supplier.

Correctness: Yes.

Clarity: Yes

Relation to Prior Work: The Bercea et al. (APPROX 2018) should be discussed.

Reproducibility: Yes

Additional Feedback: I feel your exposition would be more clear if the dependence of the frequency-distributor LP on lambda was reflected by the notation more explicitly. Typos: p. 2, line 70 "..._the_... general case" p. 4, line 142: The sentence "For any point... for it belongs..." sounds strange to me. p. 4, line 145: "...the a signature..."


Review 3

Summary and Contributions: The paper considers the k-center problem with fairness constraints: the fraction of points in each cluster coming from a single group should come within a lower and upper bound. Clustering problems with fairness constraints have received a lot of attention recently. The paper builds upon previous work and improves the approximation guarantee from 4 to 3 in the case when the potential center locations equals the points. The reported running time is also faster than prior approaches. The basic idea is simple: - Run the greedy k-center algorithm to get k-centers. - Assign the points (fractionally) to centers so as to satisfy the fairness constraints using an LP. The analysis follows along the lines of the standard k-center analysis: the points in an optimal cluster is within distance 3 from a center in our greedy solution in the case when potential centers equals the points. To speed up the the LP they also consider the notion of interchangeable points which allow them to upper bound the number of variables to be roughly the minimum of 2^k * #groups or Nk.

Strengths: Clean algorithm with seemingly good experimental performance compared to previous algorithms for the same problem. Roughly same performance as the Bera et al. algorithm but faster.

Weaknesses: - The algorithmic ideas are not very novel and not of significant interest. The analysis follows the basic analysis that the greedy algorithm is a 2-approximation for k-center. - Why do you get a 5-approximation when the potential center locations are disjoint? There the greedy algorithm is a 3-approximation so you should get that all points in an optimal cluster is within distance 4 from a point in the greedy solution, right? - Also don't your ideas apply for other clustering problems such as k-median and k-means? EDIT after author feedback: I agree with the authors that in general when F neq C, the argument gives a 5-approximation. I suppose you need F subseteq C for a 4-approximation. Thanks for the nice explanation! I also read the other reviews and my score was perhaps too harsh so I increased it slightly.

Correctness: I think it is correct.

Clarity: It is well-written.

Relation to Prior Work: Yes they explain prior work well.

Reproducibility: Yes

Additional Feedback: I think it is an OK paper but the contributions are not sufficiently interesting from a theoretical or practical perspective to be accepted to NeurIPS in my opinion. specific comments: Line 97: 3-approximation linear program. What do you mean? An LP with an integrality gap of 3 or an algorithm with an approximation guarantee of 3 with respect to the value of linear program? Line 165: NK => Nk


Review 4

Summary and Contributions: The authors design a 5-approximation algorithm for the fair k-center problem with an additive violation of 0 in expectation. The algorithm gives an approximation guarantee of 3 for a special case when F=C (C is the client set and F are the facility locations). The authors reduce the problem to a fair-assignment problem. In the fair-assignment problem, the goal is to assign the clients to a set S of k facilities such that the fair-constraints are satisfied. This reduction step is similar to the algorithm of Bera et.al.[NeurIPS 2019]. Moreover, the set S can be simply obtained by running a standard greedy 3-approximation algorithm for the k-center problem. To solve the fair-assignment problem, the authors model the problem as an LP. However, their LP formulation is different from the LP formulation of Bera et.al.[NeurIPS 2019]. The authors introduce a concept of “Joiner” of a subset of S. The joiner of a subset S’ is a set of clients that exclusively belongs to every ball of radius r that is centered at a facility in S’. The authors also define the notion of the signature of a client. The signature of a client is defined as a set of those protected classes that the client belongs to. A Joiner S' and a signature c together define a set of clients L(c,S'). All such sets L(c,S’) partitions the client set C. The important property of L(c,S’) is that in a feasible assignment two clients belonging to the same L(c,S’) can be interchanged. Based on this idea, one only needs to define at most k x_j variables per set L(c.S’). Therefore, If many clients share the same signature and joiner, then the number of variables and constraints in the LP becomes significantly less than the natural LP of the problem. Due to this, LP can be solved faster decreasing the overall running time of the algorithm. Rounding is used to obtain an integral solution from the fractional values. The clients are assigned to the facilities according to a probability distribution based on fractional values. This gives an integer solution with an expected additive violation of 0. Contributions: 1. For F = C, the authors give a 3-approximation algorithm. This is an improvement over the existing 4-approximation algorithm of Bera et.al.[NeurIPS 2019]. 2. When F and C are arbitrary sets, the authors give a 5-approximation algorithm which matches with the approximation guarantee of Bera et.al.[NeurIPS 2019]. Moreover, the authors improved the additive violation to 0 in expectation, in contrast to the previous 4*delta+3 violation (deterministic) of Bera et.al.[NeurIPS 2019]. 3. For the fair-assignment problem, the authors formulated an LP that has a less or equal number of variables and constraints than the natural LP formulation due to Bera et.al.[NeurIPS 2019]. 4. The experimental results are better than the previous results of Bera et.al.[NeurIPS 2019] and Ahmadian et.al.[KDD 2019] in terms of running time and the cost of the solution. In terms of the additive violation, the algorithm performs better on the data-sets with high delta value. I have read the author response.

Strengths: 1. The concept of joiner and signature is new and is effective since it helps in reducing the number of variables and constraints in the LP. I think this idea simple and nice. Also, it may be further used to design better algorithms for the problem. 2. The idea of randomized assignment seems nice since it gives an additive violation of 0 in expectation. Moreover, in the case of overlapping classes, it shows an improvement over the previous results theoretically and practically. 3. Experimental evaluation shows that the algorithm is faster than the previous algorithms of Bera et.al.[NeurIPS 2019] and Ahmadian et.al.[KDD 2019].

Weaknesses: Bera et.al.[NeurIPS 2019] showed that their algorithm gives a 4-approximation when C = F. However, using the analysis mentioned in this paper, the algorithm of Bera et.al.[NeurIPS 2019] also gives a 3-approximation. However, the algorithm is fast and gives low additive violations.

Correctness: Yes

Clarity: Yes

Relation to Prior Work: Yes

Reproducibility: Yes

Additional Feedback: I think the result of Theorem 1 is correct. However, the proof is not very clear. Please address the following points about the proof: 1. In line 201, how does every remaining client is at a distance at most 3.lambda from sigma(u_a)? Note that, for a remaining client, there is indeed a center in S at a distance less than 3.lambda. However, that center might not be sigma(u_a). 2. In paragraph 3, it is not clear which partition P_i is assigned to which center in S and how the fair constraints would be satisfied if multiple partitions are assigned to the same center? I think a reference to Claim 4 of Bera.et.al.[NeurIPS 2019] is required.

[Author Response · NeurIPS 2020]

We thank all the reviewers for their incisive comments and pointing out typos, all of which will be fixed in the final draft. In summary, our paper presents a new randomized algorithm for fair $k-$center/supplier clustering, together with theoretical analysis and experimental evidence. Below we address the concerns and questions from each reviewer.

**R1.** R1 questions the variance in $\mathcal{E}$, and we agree that such concern is on point. Indeed, we observe different variance in additive violations $\mathcal{E}$ on different datasets; nevertheless, $\mathcal{E} \leq 2$ is always obtained within 5 runs irrespective of the dataset. Even in the most extreme case across all experiments and datasets, the maximum additive violation observed was 8. In fact, the variance in $\mathcal{E}$ arises from fractional LP variables, but empirically almost all LP variables were integral, leading to a consistently small variance in $\mathcal{E}$. For a strictly fair comparison, we also note that Bera et al. method can be used in our algorithm to **guarantee** $\mathcal{E} \leq 4\Delta + 3$ using iterative rounding. However, we observe that the expectation guarantee was already sufficient to get few fairness violations. We will add error bars to demonstrate the robustness of $\mathcal{E}$ in the final draft. At issue is also our significantly faster runtime despite the use of binary search (BS), this is primarily attributed to the introduced joiner-signature concept. Our number of variables/constraints of the LP is very small compared to the number of points, leading to an extremely fast feasibility check in BS. Regarding $\delta$ in experiment 3, $\delta$ is a parameter defined by Bera et al. to control $\alpha_i$ and $\beta_i$, where $\beta_i = r_i(1 - \delta)$ and $\alpha_i = r_i/(1 - \delta)$, and $r_i$ is the ratio of size of group $i$ to the total number of points. Also, $\epsilon$ is used in the stopping condition $r - l < \epsilon$ during BS. The additional experiment requested is shown in Figure 2, comparing runtime with the number of protected groups on the full version of bank dataset, which has 7 protected groups in total. The results for the remaining datasets will be in the final draft. Lastly, we confirm that 30 minutes is the actual TLE, also after submission we privatized the GitHub link, which is identical to the supplementary code submitted, to maintain anonymity.

**R2.** The recommendation of enhancing the notation of the LP will be addressed in the final draft. R2 also suggests reducing overlapping groups to non-overlapping distinct groups. While the preprocessing indeed leads to distinct groups, a fair cluster for the new distinct groups does not necessarily lead to a fair cluster to the original groups. We would be happy to rerun Ahmadian et al. and Bercea et al. algorithms on the datasets with overlapping groups if there is a way. R2 also rightfully mentions that Bercea et al algorithm was not included in the experiments. The main reasons were that the algorithm used $\Theta(N^2)$ variables and constraints in the LP, which was likely to TLE on large datasets (Just like the $\Theta(N^2)$ Algorithm 2 of Ahmadian et al. did), and did not have any practical variant, such as Algorithm 4 in Ahmadian et al. The algorithm also did not have any publicly available implementation that we were aware of and did not support overlapping groups. Nonetheless, we agree and will definitely include the algorithm in the discussion, and will try to implement a practical variant of the Bercea et al. algorithm (Instead of setting $L = C$, set $L$ as $2k$ centers returned by the greedy $k - center$) and report its performance in the final camera-ready version.

**R3.** R3 questions the $5-$approx ratio for the $k-$supplier variant of the algorithm ($F \neq C$). Consider an optimal cluster $B(o_i^*, \lambda_\alpha^*)$ and the greedy centers $S$. While it is true that: $\forall x \in B(o_i^*, \lambda_\alpha^*) \cap C, \exists f \in S$ s.t $d(f, x) \leq 3\lambda_\alpha^*$, this is not sufficient to prove that the LP is feasible. To prove the LP admits a solution, it is sufficient to prove that there is a greedy center $f \in S$ such that $B(o_i^*, \lambda_\alpha^*) \cap C \subset B(f, 5\lambda_\alpha^*)$ (See Figure 1). $4\lambda_\alpha^*$ is not sufficient to guarantee this for all points. In addition, the introduced ideas of joiners and signatures used to parametrize the LP are entirely novel in this context, leading to a natural and more practical algorithm with an intuitive theoretical analysis. Furthermore, our algorithm surpassed the current state of the art both theoretically and practically by several orders of magnitude. R3 also questions if the method can be used for $k - \{means, median\}$. While it is entirely possible that this method can be adapted for those objectives (for example, define a joiner for k-median as area which has "close" nearest-neighbors at the expense of the approximation ratio), the joiner-signature method was specifically designed to parametrize fair $k - \{center, supplier\}$. Finally, line 97 should be "3-approx Algorithm using linear programming and rounding".

**R4.** R4 has two on point questions regarding the proof of Theorem 1, which we refer to as $Q1, Q2$. For $Q1$, this is indeed as pointed out incorrect, it should be that every point remaining has *some* greedy center within distance $3\lambda_\alpha^*$, however this doesn't affect the correctness of Theorem 1 (using the explanation shown above for R3) and will be fixed in the final draft. For $Q2$, we indeed implicitly used Claim 4 of Bera et al. (Neurips 19) to justify why combining a fair group leads to a fair cluster. The claim will be explicitly cited in the final draft.

Figure 1: Clarification of Theorem 1.
Blue points are points in $C$

Figure 2: $\epsilon = 0.5, \delta = 0.2$ on bank dataset. Groups 1-6 are either binary (2 categories) or tertiary (3 categories). Group 7 has 11 categories.

[Meta-Review · NeurIPS 2020]

The paper presents a speed-up of a known fair k-center/k-supplier algorithm alongside an improved approximation guarantee (4->3) for the k-center case. The latter is quite nice since there is also a lower bound of 3 for the fair (assignment) k-center problem. The authors missed the paper that contains this lower bound (Bercea et al). It is still is not tight since the proposed algorithms have a fairness violation and the lower bound is for the assignment problem, but it still seems close to the best achievable bound. However, the main contribution is the LP size reduction and resulting practical speed-up, which is demonstrated in experiments. The techniques are not surprisingly novel.